# Sustainability in Skin Care: Incorporation of Avocado Peel Extracts in Topical Formulations

**DOI:** 10.3390/molecules27061782

**Published:** 2022-03-09

**Authors:** Sara M. Ferreira, Zizina Falé, Lúcia Santos

**Affiliations:** 1LEPABE—Laboratory for Process Engineering, Environment, Biotechnology and Energy, Faculty of Engineering, University of Porto, Rua Dr. Roberto Frias, 4200-465 Porto, Portugal; up201604659@fe.up.pt (S.M.F.); zizina.fale@gmail.com (Z.F.); 2ALiCE—Associate Laboratory in Chemical Engineering, Faculty of Engineering, University of Porto, Rua Dr. Roberto Frias, 4200-465 Porto, Portugal

**Keywords:** by-products, bioactive compounds, avocado peels, natural preservatives, antioxidants, cosmetics

## Abstract

The avocado peel is an agro-industrial by-product that has exhibited a massive increase in its production in the last few years. The reuse and valorisation of this by-product are essential since its disposal raises environmental concerns. In the present study, ethanolic extracts of avocado peels of the Hass variety were obtained, for three extraction times (1.5 h, 3 h and 4 h) and analysed for their antioxidant and antibacterial properties. Antioxidant evaluations of the extracts revealed that the extraction time of 1.5 h exhibited the best results amongst the three, with a DPPH inhibition percentage of 93.92 ± 1.29 and an IC_50_ percentage, the necessary concentration of the extract to inhibit 50% of DPPH, of 37.30 ± 1.00. The antibacterial capacity of the extracts was evaluated and it was revealed that they were able to inhibit the growth and development of bacteria of the *Staphylococcus* family. The obtained extract was incorporated in two types of cosmetic formulations (oil-in-water and water-in-oil) and their stability was evaluated and compared with formulations containing synthetic preservatives (BHT and phenoxyethanol). The results of the stability evaluation suggest that the avocado peel extract has the potential to be incorporated in both types of emulsions, acting as an antioxidant and antibacterial agent, proving it to be a viable option to reduce/replace the use of synthetic preservatives. Furthermore, the avocado peel extract proved to be more effective and stable in oil-in-water emulsions. These results highlight the possibility of obtaining sustainable cosmetics, significantly reducing the negative impacts on the environment by the incorporation of extracts sourced from the avocado peel, an interesting source of phenolic compounds, an abundant and low-cost by-product.

## 1. Introduction

Personal appearance is a requirement of great importance in all segments, leading the current population to valorise more their appearance and to look for the correct tools to create the allusion of the perfect appearance imposed by society. There is a close connection between a person’s health and the use of cosmetics to beautify the skin and appear younger. Therefore, healthy-looking skin means skin that does not evidence exposure to degenerative effects, which contributes to a person being recognised as “beautiful” [1,2].

One of the most common ways to achieve beautiful and healthy skin is through the application of cosmetics. Skin is the largest organ in the human body, and it is constantly exposed to outer environmental conditions, such as pollution, and ultraviolet (UV) radiation, that can lead to extrinsic aging processes [3]. These processes are related to the emergence of wrinkles, skin dryness, loss of elasticity, and a rough-textured appearance due to the reduction of collagen, and hyaluronic acid, as well as the oxidation of certain molecules important to maintain a healthy skin appearance [4,5,6]. Therefore, it is essential to find ways to fight against them.

Skin hydration products are cosmetics whose purpose is to restore and maintain the levels of skin hydration, making it look prettier, healthier, and soft. Moisturising creams are the most common skin hydration products, with components such as humectants–compounds that have a lot of affinity with water and, therefore, attract it–and emollients –hydrophobic compounds that promote the formation of a lipid layer on the skin, preventing dehydration by occlusion [7]. Moisturising creams can be oil-in-water (O/W) or water-in-oil (W/O) emulsions, depending on which liquid is dispersed in the other. An O/W cream has water as a continuous phase, with some droplets of oil inside, whereas a W/O cream is the opposite, where the continuous phase is the oil. Figure 1 shows the difference between an O/W and W/O emulsion.

Moisturising creams are usually composed of the following ingredients: solvent (that dilutes and disperses other ingredients), emollients, humectants, emulsifiers (responsible for the stabilisation of the emulsion), thickening agents, and neutralisers [8]. Figure 2 shows the main ingredients present in cosmetics, as well as their function in the formulations, and some examples of the compounds commonly used.

Furthermore, in addition to the ingredients present in Figure 2, moisturising creams also have ingredients that act as formulation stabilisers, such as antimicrobial agents (used to prevent the growth of microorganisms, promoting product safety), chelating agents (which help to avoid the establishment of salt deposits, such as magnesium and calcium deposits), and antioxidants (preventing the oxidation of the product by delaying radical-chain reactions, and by avoiding this phenomenon they also avert the development of unpleasant smells) [9,10]. 

The cosmetic industry has been changing drastically over the past few years. Consumers express concern regarding skin aging, skin imperfections, and skin diseases, such as skin cancer and acne. These requirements have driven the investigation and innovation in this industry, and, as a response, new products are developed with the intention to help achieve pretty and healthy skin. Therefore, biologically-active ingredients–such as vitamins, enzymes, essential oils, antioxidants, and extracts of natural origin–were introduced in cosmetics. These ingredients establish new functions in cosmetics since they evidence antioxidant, antimicrobial, and antiaging properties, among others, and they are viewed as key constituents in skin protection [11,12,13]. 

Nowadays, consumers look for cosmetics mainly composed of natural ingredients and with fewer negative effects on the environment. Hereupon, researchers started to study the possibility of using the waste produced from food and plant processing to attain bioactive ingredients, mainly antioxidants, that prevent the oxidation of the product and help to reduce the oxidative damage of cellular components, and antimicrobial agents, ingredients that have antibacterial, antifungal, and antiviral properties [14,15,16]. This strategy has helped to develop greener and value-added cosmetics, as well as to reduce and recycle waste, diminishing their environmental, social, and economic impact. Moreover, this approach has in consideration all the important steps of the life cycle of the products, which follows the ideals behind the principle of circular economy [17,18]. 

Natural antioxidants and preservatives include different substances and extracts and can be achieved from a wide variety of plants, fruits, and grains, for example, green tea, grapeseed, blueberry, chestnut, and avocado. Polyphenols, flavonoids, and flavonols are some of the main natural antioxidants that can be found in plant extracts (including carotenoids and essential oils) [19,20]. This way, there is a possibility of natural antioxidants and preservatives, obtained from food and plant waste, being incorporated in cosmetics, and replacing the use of synthetic ones. Furthermore, the use of natural ingredients is also better perceived from the consumer’s point of view.

The avocado fruit, a derivative of the avocado tree of botanical name *Persea americana* Mill., is a tropical and subtropical pear-shaped climacteric berry native to Mexico and Central America. Avocado production has grown massively over the past years, making it a major agro-industrial commodity, with global productions reaching seven million tons in 2019, with Mexico being the largest avocado producer, and accounting for approximately 30% of all its production [21,22]. The majority of commercialised avocados are destined for the food industry, typically consumed as fresh fruit, as a sandwich filling, in a salad, or as sweetened desert. Processing of the pulp usually takes place to produce guacamole or avocado oil, with the avocado oil being used in the cosmetic industry, as well for the production of skin and hair care products, such as soaps and shampoos. After processing, the remnants of the fruit, namely seeds and peel that constitute approximately 33% of the fruit, are managed as solid waste, and because of their potential negative impacts on the environment, efforts are being made to promote the utilisation of these by-products [23,24,25]. 

Avocado peel is a cheap and promising option for the retrieval of phenolic compounds, and it was reported that this by-product possesses higher total phenolic content, radical scavenging capacity, and antioxidant activity compared to the pulp [26,27,28]. The majority of the phenolics present in avocado peels are flavonoids (such as catechins, procyanidins, and quercetins), and derivates of chlorogenic acid (such as caffeoylquinic acids, and coumaroylquinic acids) [29]. However, the phenolic profile of the peel varies greatly with the degree of ripeness of the avocado, as well as the country of origin and growth conditions that the fruit was subjected to [30]. 

Studies were conducted in order to characterise and evaluate the antioxidant and antimicrobial capabilities of avocado peel extracts. A study verified that different solvent extracts from the peels of two avocado varieties had a high number of phenolic compounds with elevated in vitro antioxidant potential, as well as moderate antimicrobial activity, especially towards Gram-positive bacteria. This study also confirmed the antioxidant potential of avocado extracts in a real food product [31]. Another study revealed that, in general, avocado peel ethanol extracts of different avocado varieties displayed a wide range of high-level antimicrobial activity against Gram-positive and Gram-negative bacteria [32]. Herewith, the authors suggested that avocado extracts may be used as natural additives in food products in order to extend their shelf life. In another study, the production of a tea based on avocado peels was evaluated and it was verified that phenolic and flavonoid compounds were present in the tea and that this exhibited a notable antioxidant capacity [33]. Overall, the mentioned studies indicate how the avocado peel can be used as an ingredient in cosmetic products. Up until the date of the publication of this manuscript, to the best of the authors’ knowledge, no study has been published that analyses the potential of avocado peel extract for cosmetic applications. 

In recent years, the cosmetic industry has been facing a noticeable demand for ingredients derived from natural sources [26]. Avocado by-products are a great example of agricultural waste rich in bioactive compounds that when recycled have many industrial applications [34,35]. Thereby, the purpose of this study was to incorporate avocado peel extracts into two different formulations (W/O and O/W emulsions), evaluate how they affect the performance of these formulations in terms of physicochemical and sensorial characteristics, as well as assess the possibility of the extracts to be used as natural antioxidants and antibacterial agents, allowing for the development of value-added cosmetics.

## 2. Results and Discussion

### 2.1. Extraction and Characterisation of the Avocado Peel Extract

The results of the extraction yields and antioxidant and antibacterial assays for the three extraction times evaluated are displayed in Table 1.

From the results expressed in Table 1, it was verified that the extraction time that allowed the highest yield was the 3 h extraction. A statistical analysis of variance (ANOVA) of the yield results demonstrates that the difference in the extraction yields from 3 to 4 h is not very substantial (*p* > 0.05). Thereby, it is possible to conclude that, in a compromise between time and yield, the 3 h extraction allows for better results. The yields of this study were greater than the yield values obtained in another study, for an ethanol Soxhlet extraction of avocado peels of the Hass variety, probably due to the difference in particle sizes and extraction time [36].

The DPPH inhibition percentages were very similar for the three extraction times (*p* > 0.05) but with a decrease in antioxidant capacity with increasing extraction times. The same behaviour was verified for the IC_50_ (*p* > 0.05). The 1.5 h extraction time was the one with the lowest IC_50_ value and highest DPPH inhibition percentage, hence, the one with the highest antioxidant capacity. Longer exposure times to high temperatures may have caused the loss of thermosensitive phenolic compounds, which would explain the reduction in antioxidant capacity with longer extraction times [37]. 

The results in Table 1 show that the ethanolic APE of the Hass variety developed a considerable inhibition halo diameter against the strains *Staphylococcus aureus* and *Staphylococcus epidermidis* after 24 h of incubation, whilst for the *Escherichia coli* strain, almost no halo was observed. A literature study showed that for an ethanolic extract from avocado peels of the Hass variety, the antibacterial activity for *S. epidermidis* was not verified below a 0.5 g∙L^−1^ concentration, while in the present study, the concentration of 250 g∙L^−1^ was able to inhibit bacterial growth, suggesting that *S. epidermidis* inhibition is possible with higher extract concentrations [32]. In contrast, for *E. coli*, a higher extract concentration did not result in bacterial inhibition. The literature reports show that APE exhibits antibacterial activity against *E. coli*, but with different extraction solvents, which can explain the difference in results [38,39]. The phenolic compounds procyanidin A and B (derivatives of epicatechin) are the main ones responsible for the antibacterial activity in avocado peels [40]. Longer extraction times also compromised the APE antibacterial capacity evidenced by the decrease in the inhibition halo diameter values with increased extraction times for both *Staphylococcus* species. The ANOVA demonstrates a significant difference between the inhibition halo diameters of the three extraction times (*p* < 0.05) for both *S. aureus* and *S. epidermidis*.

In view of these results, the 3 h APE displayed a satisfactory compromise between the yield and antioxidant and antimicrobial capacities and, therefore, was the selected extract to be incorporated in the W/O and O/W formulations.

### 2.2. W/O and O/W Formulations Stability 

Figure 3 displays the photos that show the appearance of the formulations before and after the accelerated thermal stability tests. 

Accelerated stability tests allow the prediction of the long-term stability of emulsions [41]. As it is possible to observe from Figure 3A,B, after the test, no phase separation nor colour change was verified for the O/W formulations. This result evidences the thermal stability of these formulations for both high and low temperatures, even for formulation F4, to which APE was incorporated as an additive. Regarding the W/O formulations, from Figure 3C,D, only formulations F1 and F2 proved to be stable at temperature changes, without colour change or phase separation being visually verified. There was a slight reduction in their viscosity to a more liquid consistency, however, it was not very considerable, suggesting that the base formulation F1 is relatively stable at temperature changes. Formulation F3, with phenoxyethanol, exhibited a slight colour change from white to a more whitish-yellow tone, and there was phase separation as well, which indicates that the addition of phenoxyethanol decreased the stability of the W/O emulsion. The same result was verified in formulation F4, which suffered a colour change, going from light green to dark green, and also exhibiting phase separation. As mentioned before, these results suggest that the base formulation is relatively stable at high and low temperatures but the incorporation of certain additives, such as phenoxyethanol and APE, can reduce the thermal stability of the emulsions. A similar result was observed in a study where a W/O emulsion with cinnamon extract was produced; the authors verified that high extract amounts could have reduced the thermal stability of their emulsions [42]. It is possible that the instability of the W/O formulations F3 and F4 is related to the thermal degradation of the additives phenoxyethanol and APE, which then destabilised the whole formulation. Therefore, microencapsulation of the APE may be needed to guarantee the temperature stability of the W/O formulation [43]. 

The results for the accelerated physical stability test can be observed in Figure 4 for all of the eight formulations analysed.

From Figure 4A, it is possible to verify that only formulations F2 and F3 displayed phase separation, while the base formulation F1 and formulation F4 did not exhibit any phase separation after centrifugation for 10 min. These results indicate that the O/W base formulation is physically stable but the addition of the synthetic additives BHT and phenoxyethanol decreased its physical stability, whereas the addition of APE did not compromise the formulation’s stability. For the W/O formulations, as shown in Figure 4B, it is possible to observe that all of them displayed phase separation. These results suggest that the W/O system produced lacks physical stability. In order to evaluate the effect of the APE in a W/O, a base formulation with physical stability is required. The stability can be improved by adding another emulsifier besides glycerol monostearate and cocamidopropyl betaine, or altering the composition of the W/O formulation.

Regarding the oxidative stability of the formulations, this parameter was evaluated with the determination of the peroxide value (PV). The PV allows the estimation of the extent of primary oxidation products in a sample, where low values imply that the oxidation extent is low. The obtained PVs for the different formulations are expressed in Table 2, as well as the respective percentage increase from day 7 to day 14. The percentage increase values from day 14 to day 21 were not included due to the non-detection of the PV values after 21 days for the F1 and F2 formulations, and due to the reduction in the PV values of formulations F4 for both the W/O and O/W emulsions.

Comparing the PVs of both formulations of F1 (no additives–negative control) and F2 (0.5% BHT–positive control) in Table 2, it is possible to verify that the oxidative state of the F1 formulations was considerably higher than those of formulation F2 for both the 7th and 14th day. This result was expected due to the presence of BHT, a commonly used antioxidant in cosmetic products to prevent their oxidation [44]. Comparing the PVs of the O/W and W/O F2 formulations, it is possible to see that the latter has a higher peroxide value, which is expected since the W/O formulation has a higher percentage of emollients (that are easily oxidised) and these belong in the continuous phase, where they are the most exposed to environmental stress. The formulations with APE presented very high peroxide values, even higher for formulation F1 of the W/O emulsion. This result may suggest that the APE increased the oxidation state of the formulations. However, it is possible that this phenomenon is the result of the presence of the phenolic compound cyanidin 3-O-glucoside, an anthocyanin that can be found in avocado peels of the Hass variety. When the avocado fruit ripens, the avocado peel changes colour from green to purple/black, and one of the causes for this is the increase in anthocyanin concentration, more specifically the accumulation of cyanidin 3-O-glucoside, which has its maximum absorbance wavelength at 516 nm [45,46]. From Equation (3) illustrated in the Methods section, the peroxide value is directly proportional to the absorbance of the sample. The wavelength used to quantify the Fe(III)-thiocyanate complex in the PV test was 500 nm, which is close to the maximum absorbance wavelength of cyanidin 3-O-glucoside. Therefore, the high absorbance values obtained in the APE formulations may be justified by the radiation absorbance of cyanidin 3-O-glucoside in these formulations. Therefore, in order to evaluate the oxidative capacity of the avocado peel by the peroxide value method, the avocado fruits need to be unripe in order to prevent cyanidin 3-O-glucoside interference, although using unripe fruit can also affect the antioxidant capacity of the extract because, as mentioned before, the phenolic profile of the avocado peel varies greatly with the degree of ripeness of the fruit.

Due to the inability to compare the PV of the formulations F4 to formulations F1 and F2, it was preferred to calculate the percentage of increase in the PV from the 7th to the 14th day. The obtained values for all six formulations are presented in Table 2. Formulations F1 were the ones with the highest increase, which was expected due to the absence of antioxidants in them. It is also possible to observe that the formulations with APE had the lowest increase in peroxide value from the 7th to the 14th day, which evidences the potential antioxidant capacity of the extract. Additionally, the increase in the PVs in the formulations with APE was lower than the increase observed for BHT, suggesting that the avocado peel extracts may possess higher antioxidant capacity than BHT over time. Figure 5 illustrates the peroxide values from day 0 to the 21st day for the O/W and W/O formulations.

Due to the non-detection of the PV values for formulations F1 and F2, the PVs of these formulations after the 21st day were considered to be zero. For both F4 formulations, the PV was calculated, due to the presence of cyanidin 3-O-glucoside in them, which absorbed radiation at 500 nm. From Figure 5 it is possible to verify a decrease in the PV for all of the formulations after 14 days and this can be explained by the fact that hydroperoxides, which are determined in the peroxide value test, are primary oxidation products that are unstable compounds and, with further propagation of the oxidation chain reaction, they reach a maximum concentration and break down into secondary oxidation products, such as aldehydes and ketones. This conversion into secondary oxidation products causes a reduction in the peroxide’s concentration [47,48]. 

Table 3 illustrates the results of the antibacterial stability assay of the formulations. For the W/O formulations, the inhibition halo diameters were measured after 24 h. For the O/W formulations, the results were obtained after 48 h, given that at the end of 24 h no inhibition halos were observed.

As it is possible to observe from Table 3, the inhibition halo diameters for all the W/O formulations were greater for *S. epidermidis* then *S. aureus*. On the other hand, an unexpected result was the antibacterial activity exhibited by the W/O F1 formulation, which had no additives, against *S. aureus* and *S. epidermidis*. A possible explanation for this result is the presence of cocamidopropyl betaine in the W/O formulation, which is an amphoteric surfactant with antimicrobial activity [49]. In Table 3, it is also possible to observe that the inhibition halos of the W/O formulations with APE (F4 and F5) were smaller than the halos of formulation F1. It seems that the presence of the APE reduced the antimicrobial activity of the formulations, but it was expected that the inhibition halos of the formulations with APE would be greater than the halos of formulation F1. The results show that the presence of APE and cocamidopropyl betaine in the same formulation decreased the antibacterial effect of the formulation. It is possible that there is an antagonistic effect when combining amphoteric surfactants, such as cocamidopropyl betaine, with phenol derivatives such as the phenolic compounds, normally found in the ethanolic APE. An alternative option to examine the antibacterial capacity of APE in the W/O formulations is, therefore, to not incorporate cocamidopropyl betaine into the formulations, and instead, incorporate a non-surfactant emulsifier to assist glycerol monostearate with the emulsification. As for *E. coli*, as expected, no halos were observed, even for the formulations F3 and F5 with phenoxyethanol (synthetic antimicrobial).

For the O/W F1 formulation, no inhibition halos were observed as expected, and, the highest inhibition halo diameter was verified against *S. epidermidis* for the O/W F5 formulation, which is a combination of the additives phenoxyethanol and APE. This inhibition diameter value is higher than the diameter value of phenoxyethanol alone, suggesting that the combination of phenoxyethanol with avocado peel extract in a cosmetic formulation, such as an O/W emulsion, has greater antibacterial capacity than with just phenoxyethanol. For *S. aureus*, the highest inhibition halos were verified for both the O/W formulations with APE (F4 and F5), higher than the inhibition halo of the formulation with just phenoxyethanol. It is important to mention that the inhibition halos for the O/W formulations were only evident after 48 h of incubation at 37 °C. This result is associated with the presence of the APE in the dispersive phase of the emulsion (oil phase), which allowed the protection and a continual release throughout the time [42]. The results suggest that, in the O/W emulsions, the APE, after long exposure times, has an antibacterial effect against the Gram-positive microorganisms *S. epidermidis* and *S. aureus*. 

Therefore, the results obtained from the stability evaluation show that the APE has the potential to be incorporated in cosmetics as an alternative to synthetic preservatives (such as BHT and phenoxyethanol), contributing to the development of more sustainable products that adhere to consumer preferences. On the other hand, the incorporation proved to be more effective and stable in the O/W formulations compared to the W/O formulations. An alternative that allows for the stable incorporation of the APE in the W/O emulsions would be the microencapsulation of the extract. 

## 3. Methods and Materials

### 3.1. Chemicals and Reagents

The avocado peel samples were acquired from a local brunch restaurant in the city of Porto. The avocados were identified as being originated from Peru and of the Hass variety. 

The extraction solvent ethanol (Ref. 1.02371.1000, C_2_H_6_O, CAS 64-17-5) was obtained from VWR (Rosny-sous-Bois, France). For the antioxidant and antimicrobial capacity, 2,2-diphenyl-1-picrylhydrazyl (DPPH) (Ref. D9132, C_18_H_12_N_5_O_6_, CAS 1898-66-4), and phenoxyethanol (Ref. PHR1008, C_8_H_8_O_6_, CAS 50-81-7) were used and purchased from Sigma Aldrich (St. Louis, MO, USA). For the moisturising formulations, the used reagents were glycerine (Ref. COSM-01216), xanthan gum (Ref. GOMA-00762), coconut oil (Ref. INCOCO-00185), beeswax (Ref. CERA-01314), shea butter (Ref. ACEI-00498) and cocamydopropryl betaine (Ref. COR-COSM-00969) acquired from GranVelada (Zaragoza, Spain), soy lecithin (Ref. L0023, CAS 8002-43-5), purchased from TCI (Tokyo, Japan), glycerol monostearate (Ref. 11324918, C_21_H_42_O_4_, CAS 31566-31-1) obtained from Thermo Fisher Scientific (Kandel, Germany), butylated hydroxytoluene (BHT) (Ref. 47168, C_15_H_24_O, CAS 128-37-0) purchased from Sigma Aldrich (St. Louis, MO, USA) and DL-Sodium lactate 50% solution obtained from VWR (Ref. 27927.298, C_3_H_5_O_3_Na, CAS 72-17-3). For the lipid oxidation tests, barium chloride dihydrate (Ref.217565, BaCl_2._2H_2_O, CAS 10326-27-9) and iron chloride III (ref. F2877, FeCl_3._7H_2_O, CAS 10025-77-1) obtained from Sigma Aldrich (St. Louis, MO, USA), iron sulphate (II) heptahydrate (Ref. 24244.232, FeSO_4_∙7H_2_O, CAS 7782-63-0) and hydrochloric acid (Ref. 20255.290, HCl, CAS 7647-01-0) were bought with VWR (Fontenay-sous-Bois, France), ammonium thiocyanate (Ref. A10632, CH_4_N_2_S, CAS 1762-95-4) acquired from Alfa Aesar (Haverhill, MA, USA) and chloroform (Ref. 438607, CH_3_Cl, CAS 67-66-3) and methanol (Ref. 414816, CH_3_OH, CAS 67-56-1) were purchased from Carlo Erba (Barcelona, Spain). A Merck Millipore Mill-Q water purification equipment, with 18.2 Ω of electric resistance (Billerica, MA, USA), used for deionised water. 

### 3.2. Methods

#### 3.2.1. Extraction of Phenolic Compounds from Avocado Peels

The avocado peels were submitted to a pre-treatment. They were washed with water and the residual water was removed with paper towel. The peels were then cut into small pieces, frozen at −80 °C and freeze-dried for 72 h. The dried samples were then milled with a coffee mill and sieved into two mesh sizes: between 50 and 100 mesh and above 100 mesh. Afterward, continuous solid–liquid extractions with a Soxhlet apparatus were performed with 220 mL of absolute ethanol for an avocado sample of 6.11 g, a 1:36 ratio [36]. The avocado samples of sizes between 50 and 100 mesh were used. In order to analyse the influence of the extraction time in the extraction yield, three extraction times were used: 1.5 h, 3 h and 4 h. The extractions were performed in triplicates. Evaporation of the ethanol was performed using a rotary vapor (BUCHI Laboratories, Flawil, Switzerland) and a gentle stream of nitrogen gas. The extraction yield was determined using Equation (1), and in order to compare the obtained yield values, a statistical one-way analysis of variance (ANOVA) was performed by calculating the *p*-value (95% confidence), with *p*-values above 0.05 indicating no significant difference between extraction times.
(1)$$\mathrm{Extraction}\;\mathrm{Yield}\;\left(\%\right)=\frac{{\mathrm{mass}}_{\mathrm{extract}}}{{\mathrm{mass}}_{\mathrm{peels}}}\times 100$$

#### 3.2.2. Characterisation of Avocado Peel Extract

In order to characterise the three extracts obtained from avocado peels, antioxidant and antimicrobial assays were performed, with an ANOVA performed to determine the significant differences between the extraction times.

The antioxidant capacity test was carried out according to the literature [50] for the three extracts. Initially, a solution of the DPPH radical at 0.2 mM in methanol was prepared by dissolving 39.4 mg of DPPH in 1 mL of methanol in an Eppendorf tube. Subsequently, the mixture was stirred vigorously using a vortex and, finally, after achieving homogeneity, it was transferred to a volumetric flask, where the sample volume was brought up to 500 mL through addition of methanol. To each well of the microplate (except wells A2 to A9 and wells in columns 10 and 11), 100 μL of methanol was added followed by the addition of 200 μL of 3 g·L^−1^ avocado peel extract solution in 2% DMSO aqueous solution, in wells A2 to A9. The sample was diluted 8 times using the microplate dilution method, where 100 µL were taken from the first row of columns 2 to 9 and transferred to the corresponding wells in the second row. The same was performed from the second row to the third and so on until reaching the last row, where 100 µL were discarded. In column 12, 100 µL of the DPPH solution was added to the first well and the dilutions were made from that well. Afterward, 100 µL of the DPPH solution was added to each well of the plate, with the exception of rows 2 and 6, where methanol was added instead. Finally, the plate was covered with the lid in order to avoid evaporation, wrapped in aluminum foil for light isolation and left to incubate for 30 min at room temperature. After the incubation period, absorbances (A) were measured at 515 nm using a microplate reader (Synergy, HT, Biotek, Winooski, VT, USA). The percentage of DPPH inhibition (% I) was calculated using Equation (2).
(2)$$\%\;I=\frac{A_{\mathrm{control}-}A_{\mathrm{sample}}}{A_{\mathrm{control}}}\times 100$$

The IC_50_ value of each extract was calculated using the linear regression equation obtained from plotting the absorbance vs. the concentration.

The antibacterial capacity of the three extracts was assessed using the disk diffusion test. Plate Count Agar, PCA, was the selected culture medium and the target microorganisms were *Staphylococcus aureus* (335 PF), *Staphylococcus epidermidis* (DSM 20044-1115-001), and *Escherichia coli* (DSM 1103). The suspensions were prepared using a pure culture and a solution of 0.9% of NaCl, whose turbidity was adjusted to 0.5 in the McFarland scale. The agar plates were inoculated with the strains and sterile disks were added to the plate and embedded with 7 μL of sample (avocado peel extract solution of 250 g·L^−1^) and controls of ultrapure water (W) as negative control; phenoxyethanol (PHEN) as a positive control. The plates were incubated for 24 h at 37 °C. Finally, the inhibition halo diameters were measured with a ruler, analysed and compared to the control samples to evaluate the antimicrobial effects of each extract [51]. All of the measurements included the disk’s diameter.

#### 3.2.3. Moisturising Cream Production

In order to evaluate the avocado peel extract (APE) antioxidant and antibacterial capacity in a cosmetic formulation, as well as its effects in the formulation’s stability, five W/O and five O/W emulsions were produced: (F1) a base formulation with no additives; (F2) with BHT (used as antioxidant positive control); (F3) with phenoxyethanol (used as antibacterial positive control); (F4) with the avocado peel extract obtained for the 3 h extraction; and (F5) with phenoxyethanol and APE. The composition of both formulations is presented in Table 4. The aqueous phase ingredients were weighed into a glass cup and heated in a hot water bath until they achieved 70 °C. The same procedure was performed for the oil phase. After both phases reached 70 °C, in the W/O formulation the aqueous phase was added to the oil phase and in the O/W formulation, the oil phase was added to the aqueous phase. These were then stirred and homogenised with a T 18 digital ULTRA-TURRAX^®^ with stirring speeds at 12,000 rpm until full homogenisation was visually verified, after 3 min. Formulations were then left to cool until room temperature, covered with paper foil and stored in a desiccator until further use and evaluation. For the formulations with additives, these were added to the formulations after cooling until room temperature and stirred again for a few minutes until homogenisation was visually observed. The percentage of additives added to the formulations is presented in Table 4, these values were added by reducing the corresponding percentage of water. 

#### 3.2.4. Stability Tests

In order to evaluate the effect of the APE in the formulations, as well as its stability, four stability tests were performed: thermal, physical, oxidative, and antibacterial stability.

##### Accelerated thermal stability test: Temperature variation

An accelerated thermal stability test was performed for all of the O/W and W/O formulations produced, with the exception of the formulations F5. The formulations were left to incubate overnight at 50 °C, followed by a 20 °C resting phase during the following day, then a low-temperature phase with an overnight incubation at 5 °C, and finally another resting phase the following day at 20 °C. Thermal stability was evaluated by visually verifying textural, consistency and colour changes [41].

##### Accelerated physical stability test: Centrifugation

For the physical stability evaluation, the formulations examined in the accelerated thermal stability test were subjected to centrifugation with a 5810 R Centrifuge by Eppendorf at 5000 rpm for 10 min. Physical stability was evaluated by visually verifying if phase separation and colour changes occurred [41]. 

##### Determination of peroxide value (PV)

The peroxide value (PV) is a parameter that helps to determine the oxidation extension of the formulations. The determination of PV and the plotting of the Fe^3+^ vs. absorption calibration curve followed a literature protocol, with slight modifications [52]. Initially, 0.1 g of sample was dissolved in 4.9 mL of a 7:3 (*v*/*v*) solution of chloroform:methanol and vortexed for 2 to 4 s. Afterward, 50 µL of an ammonium thiocyanate solution was added, and vortexed for 2 to 4 s, followed by 50 µL of an iron (II) solution. The samples were incubated for 20 min at room temperature, in the absence of light, and the absorbance was measured using a UV-Vis spectrophotometer (V-530, Jasco, OK, USA) at 500 nm. Peroxide value (PV) was calculated using Equation (3)
(3)$$\mathrm{PV}=\frac{\left(A_{s}-{\;A}_{b}\right)\times 12.26}{m_{s}\times 55.84\times 2}$$
where A_s_ refers to the sample absorbance; A_b_ refers to the absorbance of the blank; 12.26 is the value of the slope of the calibration curve Fe^3+^ vs. absorption, m_s_ corresponds to the weight of sample added (g), 55.84 is the molar mass of iron and the division by 2 gives milliequivalents of peroxides instead of milliequivalents of oxygen. To evaluate the oxidation state of the formulations with time, the peroxide value was determined after 7th, 14th, and 21st day after the production of the O/W and W/O formulations. It was assumed that the PV value at time zero was zero for every formulation. The peroxide value was determined for the formulations both F1 (negative control), F2 (antioxidant positive control) and F4 (F3 and F5 formulations were excluded, due to the absence of antioxidant additives).

##### Antibacterial assay of the formulations: Agar well diffusion method

To evaluate the antibacterial activity of the formulations (F1, F3, F4 and F5; F2 was excluded from this analysis, due to the absence of antibacterial additives), the agar well diffusion method was performed, which is similar to the disk diffusion method performed for the extracts. However, instead of small white paper disks, a small round well is made in the agar medium with the top part of a glass Pasteur pipette on the plate and the formulation is inserted into the well. These were performed for all of the ten formulations on three microorganisms *E. coli*, *S. aureus* and *S. epidermidis*. The agar plates were then left to incubate as well at 37 °C for 24 h. The diameters of the halos were measured with a ruler after 24, 48 and 72 h. All formulation samples were made in triplicates.

## 4. Conclusions

In this study, the influence of avocado peel extract in cosmetics was evaluated, producing two types of moisturising cream formulations, with oil-in-water and water-in-oil emulsions, assessing the possibility of the extract to replace synthetic preservatives to develop more sustainable cosmetics. The solid–liquid extraction, with Soxhlet, allowed a yield of 23% for an optimal extraction time of 3 h. The extracts exhibited antioxidant capacity by the DPPH method, with the highest antioxidant potential verified for the 1.5 h extraction time. The avocado peel extracts did exhibit antibacterial activity against *Staphylococcus epidermidis* and *Staphylococcus aureus* but they did not against *Escherichia coli,* in the disk diffusion method. The thermal and physical stability of the formulations were studied, where it was verified that the incorporation of avocado peel extract in the oil-in-water emulsions allowed better results than those exhibited in the water-in-oil. Regarding oxidative stability, the study revealed that over time, the avocado peel extract allowed very similar results in the antioxidant activity compared to the synthetic antioxidant BHT. Lastly, all the formulations exhibited more antibacterial activity against *S. epidermidis* compared to *S. aureus* in the agar well diffusion method, and no antibacterial activity against *E. coli* was verified. The avocado peel extract exhibited better in the oil-in-water emulsions, evidencing its potential as an additive in cosmetic formulations. Overall, this study highlighted that avocado peels can be incorporated in cosmetic formulations in the form of extracts as a replacement for synthetic additives and as a functional ingredient, creating value-added products. 

## Figures and Tables

**Figure 1 molecules-27-01782-f001:**
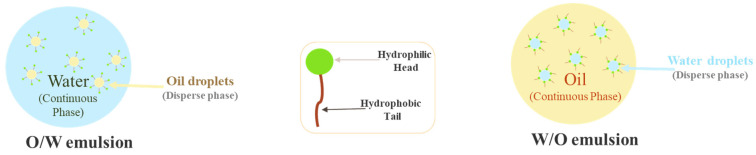
Oil in water (O/W) emulsion, and water in oil (W/O) emulsion.

**Figure 2 molecules-27-01782-f002:**
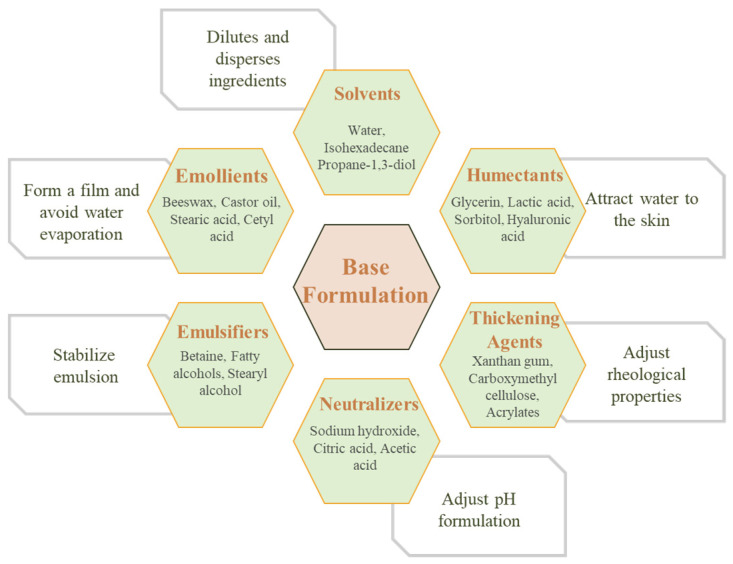
Base ingredients present in cosmetic formulations, some examples, and their function on the formulation.

**Figure 3 molecules-27-01782-f003:**
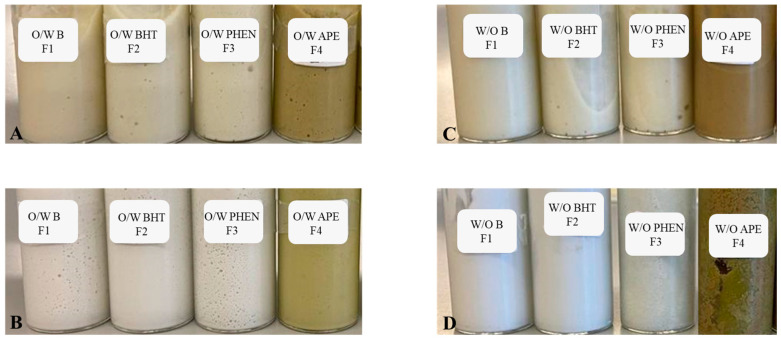
Accelerated thermal stability results of the formulations: O/W before and after the test (**A**,**B**), respectively, and W/O before and after the test (**C**,**D**), respectively. F1: base formulation without additives; F2: formulation with BHT; F3: formulation with phenoxyethanol (PHEN); F4: formulation with the avocado peel extract (APE).

**Figure 4 molecules-27-01782-f004:**
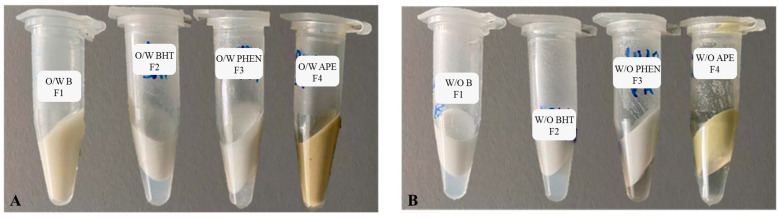
Results of the accelerated physical stability test after centrifugation at 5000 rpm for 10 min at room temperature: (**A**) O/W formulations F1 to F4; (**B**) W/O formulations F1 to F4. F1: base formulation without additives (negative control); F2: formulation with BHT (antioxidant positive control); F3: formulation with phenoxyethanol (PHEN) (antibacterial positive control); F4: formulation with the avocado peel extract (APE).

**Figure 5 molecules-27-01782-f005:**
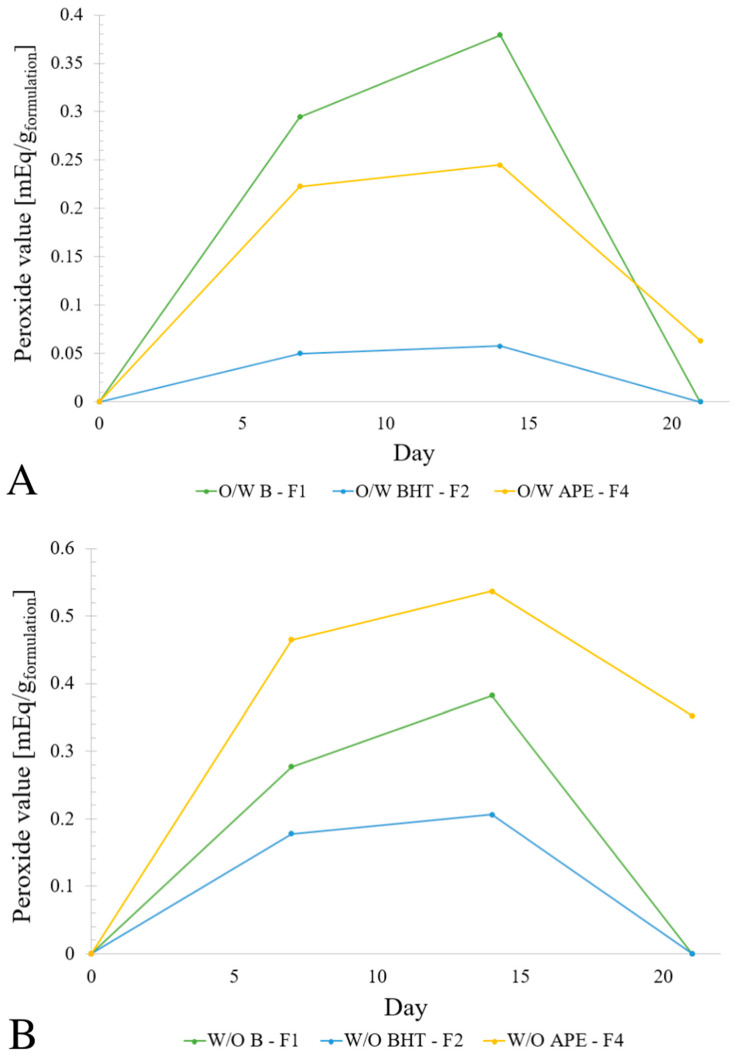
Peroxide values of the O/W and W/O formulations F1, F2 and F4 after the 7th, 14th and 21st day (**A**,**B**), respectively.

**Table 1 molecules-27-01782-t001:** Extraction yield, antioxidant, and antimicrobial capacity results for each extraction time.

	**Extraction Time**		
	1.5 h	3 h	4 h
**Yield (%)**			
	19.49 ± 0.56	22.88 ± 1.06	22.24 ± 1.80
**Antioxidant Assay**			
DPPH Inhibition (%)	93.92 ± 1.29	92.89 ± 0.77	91.18 ± 0.48
IC_50_ (µg_sample_∙mL_DPPH_^−1^)	37.30 ± 1.00	38.23 ± 2.33	38.79 ± 0.70
**Antibacterial Assay**–Inhibition halos (mm)			
*E. coli*	<5.0	<5.0	<5.0
*S. aureus*	13.0 ± 0.0	12.7 ± 0.9	10.0 ± 0.8
*S. epidermidis*	14.0 ± 0.8	12.7 ± 0.5	9.3 ± 0.5

DPPH—2,2-diphenyl-1-picrylhydrazyl; IC_50_—the necessary concentration of extract to inhibit 50% of DPPH. The results are expressed as means ± standard deviations of n = 3 independent measurements.

**Table 2 molecules-27-01782-t002:** Peroxide values after the 7th, 14th and 21st day of production for the O/W and W/O formulations F1, F2 and F4, and the percentage increase from the 7th to the 14th day.

	Peroxide Values (mEq/g_formulation_)					
	O/W Emulsion			W/O Emulsion		
Formulation	F1	F2	F4	F1	F2	F4
**7th day**	0.294 ± 0.5 × 10^−3^	0.049 ± 4.0 × 10^−3^	0.222 ± 1.3 × 10^−3^	0.277 ± 0.7 × 10^−3^	0.177 ± 3.1 × 10^−3^	0.464 ± 3.0 × 10^−3^
**14th day**	0.379 ± 2.0 × 10^−3^	0.057 ± 5.1 × 10^−3^	0.244 ± 4.3 × 10^−3^	0.382 ± 1.8 × 10^−3^	0.206 ± 3.2 × 10^−3^	0.536 ± 3.1 × 10^−3^
**21th day**	n.d.	n.d.	0.063 ± 1.6 × 10^−2^	n.d.	n.d.	0.353 ± 4.1 × 10^−3^
**PV increase from the 7th to the 14th day (%)**	29	15	10	38	16	15

F1: base formulation without additives (negative control); F2: formulation with BHT (antioxidant positive control); F4: formulation with the avocado peel extract (APE). n.d.—not detected. The results are expressed as means ± standard deviations of n = 3 independent measurements.

**Table 3 molecules-27-01782-t003:** Results of the antibacterial assay inhibition halo diameter values for the O/W and W/O formulations and pH value.

	Antibacterial Assay Inhibition Halos (mm) of the Formulations							
	O/W Emulsion				W/O Emulsion			
	F1	F3	F4	F5	F1	F3	F4	F5
** *E. coli* **	<5.0	<5.0	<5.0	<5.0	<5.0	<5.0	<5.0	<5.0
** *S. aureus* **	<5.0	16.7 ± 1.3	20.7 ± 0.5	20.7 ± 0.5	11.0 ± 0.0	12.7 ± 0.5	8.7 ± 0.5	10.3 ± 0.5
** *S. epidermidis* **	<5.0	17.3 ± 0.5	18.3 ± 1.3	22.0 ± 1.4	16.3 ± 0.9	17.0 ± 0.0	13.7 ± 0.9	15.0 ± 0.0
	**pH**							
	5.5	5.5	5.5	5.5	6	6	5.5	5.5

F1: base formulation without additives (negative control); F3: formulation with phenoxyethanol (antibacterial positive control); F4: formulation with the avocado peel extract (APE); F5: formulation with phenoxyethanol and avocado peel extract. The results are expressed as means ± standard deviations of n = 3 independent measurements.

**Table 4 molecules-27-01782-t004:** Composition of the base formulation (F1) of the two emulsions, with the respective function and percentage of the ingredients.

Emulsion	Ingredient			Function		Percentage (%)
**W/O**	**Aqueous phase**					
	Ultrapure water			Solvent		60.0
	Glycerin			Humectant		10.0
	Sodium lactate 50% solution			Humectant		2.0
	Glycerol Monostearate			Emulsifier		2.0
	Cocamidopropyl betaine			Emulsifier		4.0
	**Oil Phase**					
	Shea Butter			Emollient		14.0
	Beeswax			Emollient		8.0
**O/W**	**Aqueous phase**					
	Ultrapure water			Solvent		70.0
	Glycerin			Humectant		7.6
	Sodium lactate 50% solution			Humectant		3.2
	Xanthan Gum			Thickening Agent		0.8
	**Oil Phase**					
	Coconut oil			Emollient		7.6
	Beeswax			Emollient		6.8
	Soy lecithin			Emulsifier		2.5
	Cocamidopropyl betaine			Emulsifier		1.5
**Additives**	**Formulation**					
	F1	F2	F3		F4	F5
BHT	-	0.5	-		-	-
PHEN	-	-	1.0		-	0.5
APE	-	-	-		0.5	0.5

BHT: Butylated hydroxytoluene; PHEN: Phenoxyethanol; APE: Avocado Peel Extract.

## Data Availability

Not applicable.

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
