# Peer review of "Sustainability in Skin Care: Incorporation of Avocado Peel Extracts in Topical Formulations"

_molecules, 2022, doi:10.3390/molecules27061782_

Round 1

Reviewer 1 Report

The manuscript reports about the opportunity to incorporate the avocado peel extracts in topical formulations for the skin care. It shows a good level of data.

Authors should moderately revise the text.

The moderate check is required for the English language and style.

The last part of the final sentence of the Abstract is not appropriate 'a rich, abundant and low-cost by-product'. 'Rich' in what? Did authors study the chemical composition of it ?

May be Graphical Abstract before the Introduction should be signed as Figure 5 and it can be referenced in the text?

The sections' numbering is incorrect (for instanse, there is 1. Conclusion)

The introduction is too lengthy and includes many well-known concepts.

In the middle of Figure 1 the captions should be changed (hydrophilic tails are marked at the bottom).

line 490: number 2 should be replaced by two 'producing two types'.

The Discussion section should be expanded via comparing obtained results with the literature dates.

The extraction yield was determined in this research (line 384), not the total phenolic compounds. That is why the conclusion (lines 506-508) 'Overall, this study high lighted that avocado peels could be a good source of natural extracts rich in phenolics' is not appropriate.

Generally, phenolic conpounds of avocado are mentioned more than 15 times in the text of article but authors did not study them at all. It may be worthwhile to expand the study of the chemical compositions of it that have an effect on biological activity.

The italic type should be used everywhere for the latin names of species and second word must be in lower case, for example:

line 590 (Persea americana Mill .)

line 593 (Persea americana)

line 639 Grape (Vitis vinifera L.) etc.

Author Response

Manuscript ID: molecules-1625317

Title: Sustainability in Skin Care: Incorporation of Avocado Peel Extracts in Topical Formulations

Authors: Sara M. Ferreira, Zizina Falé, Lúcia Santos *

Section: Medicinal Chemistry

Special Issue: New Trends in Skin Care: Topical Delivery of Cosmeceutical Molecules

Note: The authors wish to express their appreciation to the Reviewers for their valuable comments on the manuscript. We hope that our revision can answer to the queries posed and may reflect an effective improvement of our work.

Answer to Reviewer's comments:

Reviewer 1 

The manuscript reports about the opportunity to incorporate the avocado peel extracts in topical formulations for the skin care. It shows a good level of data.

Answer: The authors are grateful for the comment.

Authors should moderately revise the text.

Answer: The authors are grateful for the comment, the text has been revised and alterations done.

The moderate check is required for the English language and style.

Answer: The authors thank for the comment above and tried to improve the manuscript following the reviewers’ suggestions.

The last part of the final sentence of the Abstract is not appropriate 'a rich, abundant and low-cost by-product'. 'Rich' in what? Did authors study the chemical composition of it?

Answer: The authors understand the comment and the last sentence was corrected. In this study, the chemical composition of the obtained extracts was not performed, since it was not the goal of the study and it would also make the manuscript too long. Nevertheless, in comparison with other reports in the literature with similar extraction methods and solvents to the ones used in the present study, it seems possible to conclude that the extracts obtained from this by-product is a good source to obtain phenolic compounds.

May be Graphical Abstract before the Introduction should be signed as Figure 5 and it can be referenced in the text?

Answer: The authors do not fully understand the reviewer’s comment. The authors tried to follow the journal rules regarding the graphical abstract.

The sections' numbering is incorrect (for instance, there is 1. Conclusion)

Answer: Thank you for your comment. The section numbering was corrected. The authors would like to inform the reviewers that there has been an issue with the numbering of the sections. The manuscript was submitted in a Microsoft Word format with the sections correctly numbered, but the conversion of the manuscript to the PDF format that was sent to the reviewers had the sections incorrectly numbered.

The introduction is too lengthy and includes many well-known concepts.

Answer: The authors appreciate the suggestion and a revision of the Introduction was performed, however, due to the manuscript being multidisciplinary, the authors found it important to display a complete bibliographic revision in the Introduction.

In the middle of Figure 1 the captions should be changed (hydrophilic tails are marked at the bottom).

Answer: Thank you for your comment. The image was corrected.

line 490: number 2 should be replaced by two 'producing two types'

Answer: Thank you for your comment. The suggested alteration was made.

The Discussion section should be expanded via comparing obtained results with the literature dates.

Answer: The authors appreciate the suggestion and a comparison of the yield results with those of another article was added. For the stability tests of the formulations produced, it is difficult to compare the results obtained with the literature data due to the lack of research on avocado peel extracts incorporated in topical formulations.

The extraction yield was determined in this research (line 384), not the total phenolic compounds. That is why the conclusion (lines 506-508) 'Overall, this study high lighted that avocado peels could be a good source of natural extracts rich in phenolics' is not appropriate.

Answer: The authors are grateful for the comment and an alteration of the phrase in the Conclusion section                                                           was made.

Generally, phenolic compounds of avocado are mentioned more than 15 times in the text of article but authors did not study them at all. It may be worthwhile to expand the study of the chemical compositions of it that have an effect on biological activity.

Answer:  The authors understand the comment, which is a very pertinent one, nevertheless, the authors did not perform a chromatographic analysis of the avocado peel extract obtained during the experiments due to the existence of literature data that evidence the presence of a variety of phenolic compounds in the avocado peel of the Hass variety. This literature data includes similar extraction methods and solvents to the ones used in the present study, so it was concluded that the major compounds present in the extract would be phenolics. But as mentioned in the manuscript, the percentage of these compounds may vary as it depends on the origin of the samples and other factors such as the variety of the avocado used and the growth conditions. Additionally, characterizing the extracts (with a chromatography) would make the manuscript very long, more than it already is.    

The latin names of species and second word must be in lower case, for example:

line 590 (Persea americana Mill .)

line 593 (Persea americana)

line 639 Grape (Vitis vinifera L.) etc.

Answer: As suggested, the latin names were verified and corrected.

Reviewer 2 Report

The work “Sustainability in Skin Care: Incorporation of Avocado Peel Extracts in Topical Formulations” presents a representative analysis of the proposed study, however, some changes and corrections are necessary and are described below:

INTRODUCTION

#Line 67: I suggest authors place Figure 2 below line 67.

# First and eighth paragraphs are too long.

RESULTS AND DISCUSSION

#Line 149: Section number must be 2. Check all sections and subsections.

#Line 159: A statistical analysis of the results could show whether the differences between the means are significant or not.

#Line 166: There is some reference about this sentence?

#Line 312: Why is this result expected? Does the phenolic profile of APE not show antibacterial activity?

#Line 337: Would not it be more appropriate then to measure both emulsions (O/W and W/O) after 48h?

I believe that important analyzes such as HPLC, or at least total phenolics, and emulsion viscosity could contribute to the discussion of the results.

MATERIAL AND METHODS

#Table 4: How were the formulations defined?

#Line 450: Why F5 was not analyzed?

#Line 459: What is the method reference?

#Equation 3: What does the number 12.26 in the equation mean?

#Line 475: Would not it be more correct to perform an analysis at zero time?

#Line 479: It is unclear why F2 was excluded from the analysis. In the previous item, is it not mentioned that F2 will be an antioxidant control, and even if it was why a part can not be used for antimicrobial analysis?

#Line 488: Why was no statistical analysis performed?

Author Response

Manuscript ID: molecules-1625317

Title: Sustainability in Skin Care: Incorporation of Avocado Peel Extracts in Topical Formulations

Authors: Sara M. Ferreira, Zizina Falé, Lúcia Santos *

Section: Medicinal Chemistry

Special Issue: New Trends in Skin Care: Topical Delivery of Cosmeceutical Molecules

Note: The authors wish to express their appreciation to the Reviewers for their valuable comments on the manuscript. We hope that our revision can answer to the queries posed and may reflect an effective improvement of our work.

Answer to Reviewer's comments:

Reviewer 2 

Comments and Suggestions for Authors

The work “Sustainability in Skin Care: Incorporation of Avocado Peel Extracts in Topical Formulations” presents a representative analysis of the proposed study, however, some changes and corrections are necessary and are described below:

Answer: The authors are thankful for the comments above and tried to improve the manuscript following the reviewers’ suggestions.

INTRODUCTION

 #Line 67: I suggest authors place Figure 2 below line 67.

Answer: Thank you for your comment. The figure was place accordingly to the suggestion.

# First and eighth paragraphs are too long.

Answer: Thank you for your comment. The respective paragraphs were modified.

RESULTS AND DISCUSSION

#Line 149: Section number must be 2. Check all sections and subsections.

Answer: Thank you for your comment. The suggested alteration was made.

#Line 159: A statistical analysis of the results could show whether the differences between the means are significant or not.

Answer: Thank you for your comment. The suggested statistical analysis has been made for the results of the extract characterization section.

#Line 166: There is some reference about this sentence?

Answer: Thank you for your comment. The reference was added to the manuscript.

#Line 312: Why is this result expected? Does the phenolic profile of APE not show antibacterial activity?

Answer: Thank you for your comment. The result obtained was not the expected one. The expected result was that the W/O formulations with the avocado peel extract (APE) would exhibit greater inhibition halos comparatively to the formulations without the APE. The APE does show antibacterial activity as demonstrated by the results of the disk diffusion method in the characterization of the APE section. In the W/O formulations with APE, these have antibacterial activity, however, it was lower than that activity exhibited by the formulations without the APE, which was the unexpected result.

#Line 337: Would not it be more appropriate then to measure both emulsions (O/W and W/O) after 48h?

Answer: Thank you for your comment. Both emulsions were measured after 24 and 48 hours. However, in the O/W emulsion, the halos only appeared after 48 hours; in the W/O emulsions, the halos appeared after the 24 hours and maintained their diameter after 48 hours.

I believe that important analyzes such as HPLC, or at least total phenolics, and emulsion viscosity could contribute to the discussion of the results.

Answer: The authors agree and are thankful for the comment, the suggested analysis would certainly contribute to a better understanding and discussion of the results obtained. The authors did not perform a HPLC analysis of the avocado peel extract obtained during the experiments due to the existence of literature data that evidence the presence of a variety of phenolic compounds in the avocado peel of the Hass variety. This literature data includes similar extraction methods and solvents to the ones used in the present study, so it was concluded that the major compounds present in the extract would be phenolic compounds. But as mentioned in the manuscript, the percentage of these compounds may vary as it depends on the origin of the samples and other factors such as the variety of the avocado used and the growth conditions. Additionally, characterizing the extracts (with a chromatography) would make the manuscript very long, more than it already is. The authors agree that a total phenolics analysis of the extracts and an emulsion viscosity should have been performed to better discuss the results obtained.     

MATERIAL AND METHODS

#Table 4: How were the formulations defined?

Answer: The base formulation F1 was defined as the negative control of the experiments because it has all of the ingredients in table 4, but none of the additives listed at the bottom of the table. Formulations F2 was defined as a positive antioxidant control, F3 as the positive antibacterial control and F4 and F5 as the target formulations being evaluated.

#Line 450: Why F5 was not analyzed?

Answer: The authors are grateful for the question. During the planning of the methods to be performed, formulation F5 was produced primarily to analyse only the effect of combining the APE with phenoxyethanol in the antibacterial assay, it was not intended to be analysed for its physical and thermal stability. Nonetheless, the authors did perform the accelerated physical and thermal stability tests on F5, and found that this formulation always exhibited the same results as the F3 formulation (with phenoxyethanol) in the accelerated stability tests. For example, in the O/W formulations accelerated physical stability test, the F5 formulation also exhibited phase separation like the F3 formulation with phenoxyethanol. Since the results were the same, the authors choose to exclude F5 formulation from the accelerated stability tests in order to reduce the number of results being analysed and only consider the F5 formulation in the antibacterial analysis as it was originally intended.

#Line 459: What is the method reference?

Answer: Thank you for the observation, the corresponding method reference was added to the text.

#Equation 3: What does the number 12.26 in the equation mean?

Answer: Thank you for the question. The number 12.26 is the value of the slope when plotting the Fe3+ vs absorption calibration curve, which was determined in this study. The determination of the calibration curve is elucidated in the method reference 52. Nonetheless, an explanation was added to the manuscript to make it clearer.

#Line 475: Would not it be more correct to perform an analysis at zero time?

Answer: The authors are grateful for the question, and yes, it would be more correct to perform the analysis at zero time, or, at least, 24 hours after the production of the formulations. Due to a lack of availability out of the authors’ control, it was not possible to measure the peroxide value right after the production of the formulations. It was only possible to perform these measurements a few days after the production of the formulations, so the authors considered that the peroxide value would be equal to zero for all of the formulations at time zero. This assumption was made based on the literature evidence that the peroxide value increases with time and later decreases (mentioned in the text with the references 45 and 46), and as it is possible to observe in graphs A and B from Figure 5, from the 7th to the 14th day, the Peroxide value increases, and from the 14th day and beyond, the Peroxide value decreases, therefore, from day zero to the 7th day the peroxide value should increase. But yes, as mentioned before, the authors are aware that the more correct procedure is to measure the peroxide value right after the production of the formulations.

#Line 479: It is unclear why F2 was excluded from the analysis. In the previous item, is it not mentioned that F2 will be an antioxidant control, and even if it was why a part can not be used for antimicrobial analysis?

Answer: The authors are grateful for the comment, in the previous item the information about F2 being a positive antioxidant control was added. And yes, F2 could have been used for the antibacterial analysis, but in order to reduce the number of samples being analysed in the manuscript, only the ‘’corresponding positive controls’’ were considered, that is, only the formulation with BHT was considered in the oxidation analysis and only the formulation with phenoxyethanol was considered in the antibacterial analysis.

#Line 488: Why was no statistical analysis performed?

Answer: The authors are grateful for the comment, a statistical analysis was added to the manuscript.

Round 2

Reviewer 1 Report

The sections' numbers are incorrect (for instance, there is 1. Inroduction and 1. Conclusion)

Reviewer 2 Report

The suggestions were accepted by the authors and I recommend the publication of the manuscript.